# Translating Research for the Radiotheranostics of Nanotargeted ^188^Re-Liposome

**DOI:** 10.3390/ijms22083868

**Published:** 2021-04-08

**Authors:** Chih-Hsien Chang, Ming-Cheng Chang, Ya-Jen Chang, Liang-Cheng Chen, Te-Wei Lee, Gann Ting

**Affiliations:** 1Isotope Application Division, Institute of Nuclear Energy Research, Taoyuan 32546, Taiwan; mcchang@iner.gov.tw (M.-C.C.); yjchang@iner.gov.tw (Y.-J.C.); lcchen@iner.gov.tw (L.-C.C.); tewei123456@gmail.com (T.-W.L.); gann.ting@msa.hinet.net (G.T.); 2Department of Biomedical Imaging and Radiological Sciences, National Yang Ming Chiao Tung University, Taipei 11221, Taiwan

**Keywords:** liposome, nanoliposome, nuclear imaging, radiotheranostics, rhenium-188

## Abstract

Nanoliposomes are one of the leading potential nano drug delivery systems capable of targeting chemotherapeutics to tumor sites because of their passive nano-targeting capability through the enhanced permeability and retention (EPR) effect for cancer patients. Recent advances in nano-delivery systems have inspired the development of a wide range of nanotargeted materials and strategies for applications in preclinical and clinical usage in the cancer field. Nanotargeted ^188^Re-liposome is a unique internal passive radiotheranostic agent for nuclear imaging and radiotherapeutic applications in various types of cancer. This article reviews and summarizes our multi-institute, multidiscipline, and multi-functional studied results and achievements in the research and development of nanotargeted ^188^Re-liposome from preclinical cells and animal models to translational clinical investigations, including radionuclide nanoliposome formulation, targeted nuclear imaging, biodistribution, pharmacokinetics, radiation dosimetry, radiation tumor killing effects in animal models, nanotargeted radionuclide and radio/chemo-combination therapeutic effects, and acute toxicity in various tumor animal models. The systemic preclinical and clinical studied results suggest ^188^Re-liposome is feasible and promising for in vivo passive nanotargeted radionuclide theranostics in future cancer care applications.

## 1. Introduction

Over the past few decades, theranostics has emerged as a field in which diagnosis and targeted therapy are combined to achieve a personalized treatment approach to the patient [1]. The progress in molecular biology and the understanding of malignant transformation and tumorigenesis have revealed recognition of the widespread applicability of the hallmarks of cancer concepts. These novel concepts increasingly affect the development of new means to treat human cancer in personalized and translational medicine. Conventionally, anticancer drugs are used to systemically destroy any residual or metastatic tumor cells, but they cannot discriminate between neoplastic and non-neoplastic cells [2]. Besides, poor solubility and distribution, unfavorable pharmacokinetics, and high-tissue damage or toxicity are also noted in various therapeutic agents. Therefore, an effective therapeutic strategy targeting specific mechanisms involved in tumorigenesis is needed [3]. The biggest challenges in cancer diagnostics and therapeutic radionuclide agents in clinical applications include low drug bioavailability within cancer cells and the high toxicities to normal organs [4]. An ideal delivery system can carry radionuclides into the cancer targeting area and releases the desired cytotoxic concentration at the right time. The importance of nanotechnology to radionuclide delivery systems in molecular imaging and targeted radiotherapy has been recognized this decade [5,6,7]. Several innovative radionuclides and drug delivery systems focused on cancer nanotechnology have been developed in recent investigations. Liposomes have been applied to improve the targeting of radionuclide transport into tumor lesions [4,8,9,10,11,12]. Besides, nanotargeting delivery systems have also been revealed to reduce radionuclide cytotoxic agent side-effects.

Liposome is one of nanomedicine formulations originally designed to improve the distribution and target site accumulation of systemically administered therapeutic agents [4,12]. They are spherical, self-closed formed lipid bilayers with phospholipids in which they are entrapped radionuclides [7]. These nanosized drug delivery systems are popularly used because of increased absorption, delayed excretion, decreased uptake and removal from circulation by the reticuloendothelial system, longer half-life within the blood circulation and lower toxicity. Besides, liposomes can accumulate in tumors through leaky tumor vasculature and the enhanced permeability and retention (EPR) effect [2,13]. With these outstanding properties, liposomes have been used as targeted drug delivery systems. Recent liposome concomitant imaging radionuclides and therapeutics have drawn attention to an approach combining diagnostics and treatment in various studies [8,14,15]. Successfully combining molecular imaging and nanomedicine approaches has fundamentally contributed to a new and highly interdisciplinary research field. Harrington and his colleague reported the biodistribution and imaging of ^111^In-diethylenetriaminepentaacetic acid (DTPA)-labeled PEGylated (polyethylene glycol, PEG) liposomes in advanced cancer patients. Tumor images from various cancer types including the breast, head and neck, bronchus, glioma and cervix were obtained by gamma camera and single photon emission computed tomography/X-ray Computed Tomography (SPECT/CT) [16]. Recently, with improvements in radiolabeling methodologies development [17], PEGylated technology [18], scanner technology and imaging quality, the liposomes in cancer diagnosis and therapy can be potentially used in clinical applications. The ^186^Re-liposome was developed by a remote-loading method for ^186^Re encapsulated in liposome [19]. Liposomes in the 100-nanometer size range have been the most investigated carrier for convection enhanced delivery (CED) drug delivery to the brain. They have been utilized as a carrier for radiotherapeutic rhenium-186 radionuclides to very high levels of specific activity for treating glioblastoma [9]. The phase I/II clinical studies including maximum tolerated dose (MTD), safety and efficacy of rhenium nanoliposomes in recurrent glioma of ^186^Re-liposome (ReSPECT^TM^) have been promoted in the United States by Plus Therapeutics, Inc. The development of personalized theranostic radiophamaceuticals show greater accuracy in selecting patients who may respond to treatment and allow for the assessment of the therapeutic response [7]. Rhenium (Re) and technetium (Tc) belong to the manganese family (VIIB). The chemical properties of ^188^Re and ^99m^Tc are highly similar to each other. With the readily detectable gamma ray properties and short physical/biological half-life, ^99m^Tc has become the one of the most widely used radioisotopes for radiopharmaceuticals. The major advantage of ^188^Re over other radionuclides in internal radiotherapy application is the availability and cost-effectiveness of the ^188^W/^188^Re-generator (T_1/2_ = 69.4 days), which has a longer shelf-life and a higher specific activity. This is convenient to use and easier to continuously supply to remote areas of the world. There are two major advantages of ^188^Re over other radionuclides in radio-theranostic application. One is the radio-therapeutic effect of ^188^Re derives from the high energy E_β(max)_ value of 2.12 MeV (71.1%), allowing a maximum tissue penetration range R_β(max)_ of 10.4 mm. The other is the 155-keV (15.6%) gamma ray with a half-life of 16.7 h allows tumor targeting to be monitored using gamma ray nuclear imaging.

Ionizing radiation therapy is used as a primary treatment for many types of cancer. Since 1990, radiation-induced apoptosis has been observed in several animal tumors and in different cell lines, and many investigators have identified molecular signals that control the induction of programmed cell death (apoptosis) in cells exposed to radiation [20]. Although apoptosis pathways induced by ^188^Re beta-irradiation have been reported [21], the molecular mechanism is still an unmet need. We demonstrated the colon tumors from treated mice had a 26-fold increase in the numbers of apoptotic cells compared with those from the normal saline control mice at 8 h after treatment with ^188^Re-liposome [22]. ^188^Re-liposome significantly reactivated the p53 pathway, suppressing the epithelial-to-mesenchymal transition (EMT) process [23,24] and the reversal of glycolysis in an ovarian tumor model [23]. ^188^Re-liposome also effectively suppressed the expression of stemness-relevant markers of cancer-stem cells and may be a novel treatment for ovarian cancer when delivered intraperitoneally [23]. In addition, ^188^Re-liposome could induce the autophagy/mitophagy of cancer stem cells, leading to tumor regression in the ovarian tumor animal model [25].

MicroRNA, an abundant family of short (19–25 nucleotides) noncoding RNAs, can modulate gene expression upon binding to target mRNAs. The aberrant expression of miRNAs can regulate diverse cellular processes, including the apoptosis of tumor cells [26]. Microarray array analysis revealed the tumor suppressor microRNA let-7 could be induced by ^188^Re-liposome to regulate downstream genes in human head and neck squamous cell carcinoma (HNSCC) [27]. Most ^188^Re-liposome that up-regulate microRNAs, including miR-206-3p, were categorized as tumor suppressors, while down-regulate microRNAs, including miR-142-5P, were oncogenic [28]. Studies on the regulation of microRNAs correlate to the therapeutic efficacy of ^188^Re-liposome in HNSCC tumor [27,28]. The evidence indicated ^188^Re-liposome enhances various anti-tumor mechanisms in tumor therapy. Figure 1 overviews the possible mechanisms in the anticancer effects of ^188^Re-liposome.

Our preclinical cell and animal model investigations demonstrated the efficacy and safety of nanotargeted radiotheranostic ^188^Re-liposome or radio/chemotheranostic ^188^Re- doxorubicin (DXR)-liposome, which includes tumor site targeting and nuclear imaging, pharmacokinetics, biodistribution, dosimetry, the radiation killing effect, and synergistic radio/chemo-combination effects. We also completed a phase 0 clinical trial for 12 patients, which showed the potential of ^188^Re-liposome as a new nanotargeted anti-cancer radiotherapeutic [29]. The current review and summaries focus on the systemic overview and comparisons of our preclinical nanotargeted ^188^Re-liposome pharmacology, therapeutic effects, and safety studies in animal models, leading to translational clinical investigations’ achievements (Table 1) and promoting the advancement and utilization of nanotargeted ^188^Re-liposome in future cancer clinical research and practical applications.

## 2. Rhenium-188 Production

The most important achievement in using ^188^Re is the development of a ^188^W/^188^Re-generator. The ^188^W/^188^Re-generator is easy to manipulate and practical, especially in some remote areas where it is difficult to transport short-living radionuclides routinely. Originally, ^188^W is produced by the ^186^W (2n,γ)^188^W nuclear reaction [49,50]. This reaction involves two successive neutron captures, making the probability of a reaction proportional to the square of the neutron flux. Consequently, the production depends on the neutron flux. However, only very high neutron flux (10^15^/cm^2^/second) can produce sufficient ^188^W. Currently, a ^188^W/^188^Re-generator design is based on ^188^Re separation from ^188^W using an alumina column. For instance, the parent nuclide, ^188^W, is loaded on the column and a ^188^Re daughter radionuclide is eluted with normal saline. The half-life of ^188^W is 69 days. Therefore, the shelf life of an ^188^W/^188^Re-generator can be longer than four months, based on the activity levels of ^188^Re required. The interval of ^188^Re elution to achieve the maximum radioactivity is approximately three days [51]. For the preparation of ^188^Re-liposome, ^188^Re could be milked from a homemade ^188^W/^188^Re-generator of the Institute of Nuclear Energy Research [52] or a commercialized Good Manufacturing Practice (GMP)/Pharmaceutical-Grade ^188^W/^188^Re generator by IRE (Institut National des Radioelements, Fleurus, Belgium).

## 3. Preparation and Characterization of ^188^Re-Liposome

^188^Re-liposome is composed of liposome (Carrier), *N*,*N*-bis(2-mercaptoethyl)- *N*′,*N*′-diethyl-ethylenediamine (BMEDA, chelator) and ^188^Re (Radionuclide) by a remote-loading method [19]. BMEDA is a nitrogen and sulfur donor (SNS) pattern ligand with a tridentate structure that has one nitrogen and two sulfur atoms (Figure 2A). These three atoms are able to offer electrons to ^186^Re, ^188^Re and ^99^Tc for organizing a lipophilic complex in a neutral state [19] (Figure 2B). The SNS/S complexes have a neutral core coordinate structure which can cross the lipophilic double membrane of a liposome. Carrier-free ^188^Re-perrhenate (NaReO_4_) and BMEDA were used to form ^188^Re-BMEDA [9]. The labeling efficiency of ^188^Re-BMEDA is usually more than 95%. Liposomes encapsulating (NH_4_)_2_SO_4_ have been chosen as the drug delivery scaffold [9]. With ^188^Re-BMEDA crossing the lipophilic bilayer of liposome, the lipophilic/hydrophilic characterization can trigger a gel-like formation and anions trapping (Figure 2C). The ^188^Re-liposome was separated from free ^188^Re-BMEDA using a PD-10 column eluted with normal saline. The overall yield of ^188^Re-liposome is about 70%. For clinical trials, ^188^Re-liposome was further filtered with a 0.22 micrometer filter in our Pharmaceutical Inspection Convention and Pharmaceutical Inspection Cooperation Scheme (PIC/S) GMP radiopharmaceutical production center. The specification for the concentration of phospholipid were 3~6 µmol/mL, a particle size of 80~100 nm, zeta potential of −3~2 mV and radiochemical purity of more than 90%. Figure 3 shows the scheme for the preparation of ^188^Re-liposomes [32]. Studies have indicated ^188^Re-liposome is stable in normal saline, serum and plasma [30,32,53]. The results from in vitro and in vivo studies indicated that about 92–98% of ^188^Re-liposomes maintain high radiochemical purity (RCP) for up to 72 h of incubation [30,31,32,53]. The evidence indicates ^188^Re-liposome has high radiochemical purity and drug encapsulation stability.

## 4. Preclinical Studies of ^188^Re-Liposome

### 4.1. Pharmacokinetics and Biodistribution

For pharmacokinetics, whole blood samples were collected for up to 168 h (lung metastatic model up to 96 h) post administration through the tail vein. Radioactivity in whole blood samples was normalized to the percentage of injected dose per gram at various time points following the intravenous injection of ^188^Re-liposome. Pharmacokinetic parameters were determined using the WinNonlin software v5.0.1 (Pharsight Corp., Mountain View, CA, USA). Non-compartmental analysis was used with the log/linear trapezoidal rule. The pharmacokinetic parameters, including time to achieve maximum concentration (Tmax), maximum concentration (Cmax), clearance (Cl), area under the tissue concentration–time curve (AUC), and mean residence time (MRT) were determined. Pharmacokinetic parameters of ^188^Re-liposome from blood in various cancer-bearing mice models have been estimated [30,32,36,37]. The elimination half-life (T_1/2λz_) of ^188^Re-liposome and ^188^Re-BMEDA was 185.33 h and 34.72 h, respectively, in a C26 ascites tumor model [32]. T_1/2λz_ of ^188^Re-liposome was 5.3-fold longer than that of ^188^Re-BMEDA in blood. The AUC_(0→∞)_ of ^188^Re-liposome and ^188^Re-BMEDA was 763.03% ID/g × h and 81.28% ID/g × h, respectively. The AUC_(0→∞)_ of ^188^Re-liposome in blood was 9.4-fold larger than that of ^188^Re-BMEDA. The total body clearance of ^188^Re-BMEDA (1.23 mL/h) was higher than that of ^188^Re-liposome (0.13 mL/h). Furthermore, The AUC_(0→∞)_ of ^188^Re-liposome and ^188^Re-BMEDA was 800.6% ID/g × h and 172.6% ID/g × h, respectively, in a C26 solid tumor model [30]. The AUC_(0→∞)_ of ^188^Re-liposome in blood was 4.65-fold larger than that of ^188^Re-BMEDA. The AUC_(0→∞)_ of ^188^Re-liposome and ^188^Re-BMEDA was 488.97% ID/g × h and 62.73% ID/g × h, respectively, in an NCI-H292 solid tumor model [40]. The AUC_(0→∞)_ of ^188^Re-liposome in blood was 7.8-fold larger than that of ^188^Re-BMEDA. These pharmacokinetic parameters indicate ^188^Re-liposome has significantly greater MRT with longer T_1/2λz_ and the greater AUC compared with those from ^188^Re-BMEDA.

All biodistribution showed ^188^Re-liposome could accumulate significant amounts in the tumor, liver, spleen and kidneys [30,32,36,40,54]. Liposome could be captured in the liver and spleen by the reticuloendothelial system [2]. The highest uptake of ^188^Re-liposome in tumors was found at 24 h after administration, and was steadily maintained until 72 h after administration. The highest tumor to muscle ratio (Tu/Mu) was up to 25.8 and 14.4 at 24 h in the C26 murine colon ascites model [36] and HT-29 human colorectal carcinoma solid tumor model [31], respectively. Very low uptake of ^188^Re-liposome has been reported in the organs of the central nervous and musculoskeletal systems [30,36]. Biodistribution studies suggested that ^188^Re-liposome exhibits high retention in blood circulation and tumor accumulation in vivo. Our study also showed ^188^Re-liposome could accumulate in brain tumors. The mechanism may be due to the neoangiogenesis properties of glioma, with newly formed vasculatures overpassing the blood–brain barrier (BBB). Besides, glioma cells can degrade tight junctions by secreting soluble factors, leading to BBB disruption [33]. The excretion of ^188^Re-liposome has been studied [36,37]. The fraction of ^188^Re-liposome excreted in the urinary and gastrointestinal tract is derived from accumulative urine and feces using metabolic cages [37,55]. The total excreted fraction of ^188^Re-liposome by urine and feces was about 47–87% in different tumor models. The ^188^Re-liposome was mostly excreted via feces (29–61%), suggesting the importance of hepatobiliary excretion for these compounds.

### 4.2. Longitudinal MicroSPECT/CT Imaging of Tumor Targeting for ^188^Re-Liposome

Several tumor-bearing mouse models demonstrated longitudinal microSPECT/CT could be used as the imaging of tumor targeting by using ^188^Re-liposome [30,32,34,35,38]. The accumulation and localization of nanotargeted ^188^Re-liposomes has been studied in C26 solid mouse models [30,34]. The images revealed ^188^Re-liposomes (Figure 4B) and ^188^Re-DXR-liposomes (Figure 4C) remained in the tumor for up to 72 h post injection, while the corresponding images for free ^188^Re-BMEDA revealed it could not accumulate in the tumor at 4 h post injection (Figure 4A). Although uptake of ^188^Re-liposome in the spleen and liver are the common features of nanoparticles following intravenous injection into mice, the images revealed a higher uptake in tumors up to 72 h after intravenous injection. A positive correlation (r = 0.663) of tumor uptake of ^188^Re-liposome by biodistribution and microSPECT semi-quantification image analysis was obtained using Pearson correlation analysis (Figure 4D) [30]. MicroSPECT/CT imaging also revealed a higher uptake of ^188^Re-liposome in tumors up to 72 h post injection in a human LS-174T-colorectal cancer tumor-bearing mice model [38]. Whole-body autoradiography results also confirmed the correlation between microSPECT/CT imaging and biodistribution data [38]. Our results indicated molecular imaging is a non-invasive imaging modality that can longitudinally monitor the behavior of ^188^Re-liposome radiotherapeutics in the same animal across different time points.

### 4.3. Dosimetry of ^188^Re-Liposome

Radiation dosimetry analysis is an important aspect of evaluating the safety and efficacy of internal emitters for radionuclide therapy [56]. Regulatory agencies (e.g., the US Food and Drug Administration, FDA) approved the use of animal models to estimate the dosimetry before clinical trials [57]. Radiation dosimetry was conducted to estimate the absorbed doses in non-target organs, as well as in target tumors for ^188^Re-liposome, to extrapolate the animal data to humans. Radiation dose estimates for normal tissues and tumors were calculated using the OLINDA/EXM program [58].

Various studies from colon carcinoma tumor-bearing mice models indicate the absorbed dose of ^188^Re-liposome in liver, kidneys and red marrow are all about 0.24–0.40, 0.09–0.20 and 0.033–0.050 mGy/MBq, respectively [36,37,38,55]. Either liver or red marrow may be the dose-limiting organ for radionuclide PEGylated nanoliposome therapy [59]. High absorbed doses are also observed in the gastrointestinal tract, including the lower large intestine, upper large intestine and small intestine. This suggests ^188^Re-liposome was mostly excreted via feces. Very low absorbed doses of ^188^Re-liposome are noted in the organs of the central nervous and musculoskeletal systems. Bone marrow toxicity is also generally dose-limiting for β-emitting radiopharmaceuticals. Those results indicated the nanoliposome formulation did not cause higher absorbed doses in normal tissue than nontargeted formulation, but it did cause higher absorbed doses in tumors. Systemically targeted radionuclide therapy using ^188^Re-liposome is feasible, promising, and may have the advantage and benefit of reduced toxicity and improved therapeutic efficacy.

### 4.4. Acute Toxicity of BMEDA and ^188^Re-Liposome

Non-clinical safety studies are necessary to assess the safety and toxicity profiles of drug compounds under development and before clinical trials [60]. Acute toxicity studies in various animal models aiming to provide pharmaceutical information intended for human use, and the information is useful for providing preliminary identification of target organs of toxicity. Since BMEDA (C_10_H_24_N_2_S_2_) is a new chemical entity for chelating ^188^Re in ^188^Re-liposome, the acute toxicity studies of BMEDA should be evaluated. The LD_50_ value of BMEDA is estimated to be 8.13 and 8.68 mg/kg in rodents for male and female mice, respectively. No difference in body weights and no observable gross lesions are observed among 3- and 6-mg/kg BMEDA-treated and control mice [45]. Extended acute toxicity studies of BMEDA have also been confirmed in non-rodents, with beagles receiving a BMEDA dose of 1 mg/kg that had no adverse effect and no doses causing life-threatening toxicity [46].

A 28-day extended acute toxicity study for ^188^Re-liposome was performed in Sprague Dawley rats via a single intravenous injection, obtaining the “no observed adverse effect level (NOAEL)” estimated to be greater than 185 MBq (5 mCi) per rat (weight of a rat estimated to be 200 g) [48]. None of the rats died and no clinical sign was observed during the 28-day study period. No differences in the biochemistry parameters, gross lesion and histopathological damage were found between the ^188^Re-liposome treated and control groups.

### 4.5. Therapeutic and Combination Effects of ^188^Re-Liposome

The treatment of colorectal cancer patients remains a challenging problem [61]. We demonstrated the therapeutic efficacy of ^188^Re-liposome in C26 [22] or LS-174 [38] colorectal tumor bearing mice. Mice receiving various doses of ^188^Re-liposome (from 22.2 to 37 MBq) can significantly increase overall survival time by more than 60% compared with control saline injection. The results showed the dose-dependent therapeutic efficacy and prolonged survival time of ^188^Re-liposome in colorectal tumor-bearing mice [22,38]. Liposomal drugs such as pegylated liposomal doxorubicin (DXR) can be designed to improve the pharmacological and therapeutic index for cancer therapeutics [2,13], but the limited distribution of doxorubicin in solid tumors causes drug resistance and a lower chemotherapy response [62]. The developments for improving therapeutic efficacy, reducing side effects and overcoming the drug resistance of multiplex nanoliposomes are of considerable interest.

For the therapeutic efficacy in a C26 murine colon carcinoma solid tumor e model [34], the tumor volume inhibition and the survival curves following various treatments are shown in Figure 5A,B, respectively. Two (25%) of the mice treated with ^188^Re-DXR-liposome (Figure 5C) or ^188^Re-liposome were completely cured after 120 days. The bimodality radio-chemotherapeutics of ^188^Re-DXR-liposome and radiotherapeutic of ^188^Re-liposome showed better mean growth inhibition (MGI) rates (MGI = 0.048; 74 days) and (MGI = 0.134; 60 days), respectively, than those treated with chemotherapeutics of Lipo-DOX (MGI = 0.413; 38 days) (Figure 5D). The therapeutic efficacy of ^188^Re-liposome and the synergistic effect of the combination of ^188^Re-DXR-liposome point to the potential benefit and promise of the co-delivery of nanoliposome radio-chemotherapeutics for adjuvant cancer treatment on oncology applications [34].

For several decades, 5-Fluorouracil (5-FU)-based regimens have been the first choice of the primary or adjuvant chemotherapy for colorectal cancer [63,64]. We compared the therapeutic efficacy by single equivalent dose (80% maximum tolerance dose, MTD) of ^188^Re-liposome (80% MTD; 29.6 MBq) with that of 5-FU (80% MTD; 144 mg/kg) in different tumor models [22,36,37,38]. All therapeutic results including median survival time and tumor volume inhibition demonstrated better therapeutic ^188^Re-liposome efficacy (80% MTD) than that of 5-FU (80% MTD) treatment in mice. For the comparative therapeutic study of ^188^Re-liposome and 5-FU in a C26 murine colon carcinoma solid tumor mice model [22], the tumor volume inhibition and the survival curves following various treatments are shown in Figure 6A,B, respectively. With respect to therapeutic efficacy, the tumor-bearing mice treated with ^188^Re-liposome showed a better mean tumor growth inhibition rate (MGI) and longer median survival time (MGI = 0.140; 80 days) than those treated with the anti-cancer drug 5-FU (MGI = 0.195; 69 days) and untreated control mice (48 days) (Figure 6C).

In the comparative therapeutic study of ^188^Re-liposome and 5-FU in a C26 murine colon peritoneal ascites and tumor mice model [36], significant inhibitions of hemorrhagic ascites formation (Figure 7A) and the inhibition of tumor growth (Figure 7B) were observed in the ^188^Re-liposome treated group (83.3%, 63%) in comparison with the 5-FU (44.9%, 9.1%). The survival curves after various treatments were showed in Figure 7C. With respect to therapeutic efficacy, the tumor-bearing mice treated with ^188^Re-liposome showed better mean survival time and increased life span (32.8 days, 34.6%) than those treated with the anti-cancer drug 5-FU (26.7 days, 9.6%) and untreated control mice (24.3 days) (Figure 7D).

We also worked on studies of combination therapy using ^188^Re-liposome and chemotherapeutic agents. ^188^Re-liposome was combined with sorafenib (Nexavar) [42], or Lipo-Dox [43] in C26 colorectal tumor mouse models. Hsu showed mice treated with ^188^Re-liposome and Lipo-Dox pretreatment experienced limited tumor growth (Figure 8A). Seventy-five percent of the mice could survive with the combination of ^188^Re-liposome and Lipo-Dox. Groups with Lipo-Dox or ^188^Re-liposome treatment showed a survival rate of less than 25% at the same time point [43]. With respect to therapeutic efficacy, the tumor-bearing mice treated with ^188^Re-liposome and pretreated with Lipo-Dox showed increased mean survival time and increased life span (more than 120 days, 242.9%) compared to those treated with ^188^Re-liposome (57 days, 62.9%) or Lipo-Dox (47.5 days; 35.7%) and untreated control mice (35 days) (Figure 8B). ^188^Re-liposome combined with Nexavar achieved higher survival rates (75%) compared with the ^188^Re-liposome (62.5%) or Nexavar (0%) alone groups at the end of the study (Figure 8D) [42]. The therapeutic effect of the chemotherapeutic agent for the tumor significantly improved by ^188^Re-liposome demonstrated its delivery and effectiveness.

External beam radiotherapy (EBRT) can deliver high-energy radiation beams to cover both gross tumors and potential microscopic tumor cells in the vicinity of a tumor, including esophageal cancers. Our studied showed ^188^Re-liposome combined with EBRT achieved higher tumor growth inhibition (53%) compared with the ^188^Re-liposome (25%) or EBRT (30%) alone groups at 21 days post-injection (Figure 8C) [44]. Combination treatment had no additive adverse effects or significant biological toxicities on white blood cell counts, body weight, or liver and renal functions. According to combination studies of ^188^Re-liposome with chemotherapeutics or EBRT, the combination strategy may be a potential treatment modality for cancers.

## 5. Clinical Studies of ^188^Re-Liposome

The FDA guidelines for an exploratory IND (eIND) published in 2006 [65] give the ability to conduct a human trial for obtaining early pharmacokinetic and pharmacodynamics information according to the microdosing concept. The term microdosing is defined as less than 1/100th of the dose calculated to yield a pharmacological effect of the drug candidate based on primary in vitro and in vivo data and administered at a maximum dose of ≤100 µg [24,66]. Due to the BMEDA being the first to be used in humans, the eIND is an approach to evaluate the imaging, pharmacokinetic and safety of ^188^Re-liposome.

An open-label, single-arm, phase 0 clinical trial with a 111 MBq microdose of ^188^Re-liposome injected in patients with metastatic cancer was carried out. The study proved the safety of ^188^Re-1iposome in 12 patients refractory to current standard/available therapies. None of the subjects showed any serious adverse effects following ^188^Re-liposome injection, and no clinically significant abnormalities in physical exams or laboratory exams were found [29]. Patient 14, who had a nasopharyngeal tumor (NPC) over the left nasal cavity and lung metastases, showed high uptake (3.99% IA/kg) at the targeted area at 24 h after injection with ^188^Re-liposome, corresponding to the soft tissue lesion (blue arrow) seen on the corresponding MRI (Figure 9A,B). Anterior planar images of Patient 14 were obtained at different time points following injection with ^188^Re-liposome and reveal a discernible uptake of radioactivity in the left nasopharyngeal region (Figure 9C). This nasopharyngeal tumor shows a clear response to the low radioactivity of ^188^Re-liposome, as verified by nasopharyngoscopic examinations at two and twelve months after injection (Figure 9D–F) [29].

Two recurrent ovarian cancer patients received cohort 1 therapeutic activity of ^188^Re-liposome in phase 1 clinical trials. Four months after the treatment, the CA-125 started to decline. Chest film follow-up also showed a partial resolution of right pleural effusion [25]. These results suggest ^188^Re-liposome may reverse from a drug resistance status to a drug sensitive status and may be a novel strategy in overcoming drug resistance in ovarian cancer. Although the two cases do not prove the therapeutic efficacy of ^188^Re-1iposome, the extended life after treatment of the two cases is more than expected.

## 6. Conclusions

The advantages of a longer shelf-life radionuclide generator, higher energy β-emitter for radiotherapeutics and suitable gamma ray energy and half-life for nuclear imaging provide ^188^Re with suitable characteristics for internal radiotheranostics application. However, the main disadvantage of ^188^Re nuclear imaging is the presence of a high-energy background signals (478 keV, 633 keV, and 829 keV), with high-energy ultra-high resolution multi-pinhole collimators being required to optimize both image quality and quantitative accuracy for ^188^Re SPECT nuclear imaging in diagnostic applications.

The effectiveness and safety of personalized medicine involves multiple factors that contribute to cancer disease types and development, and resistance to radio/chemotherapy. We demonstrated nanoliposome is a suitable passive delivery system for ^188^Re radionuclide in various cells and tumor animal models, which include prolonging the radioactivity of ^188^Re in tumor sites with longer biologic half-life through EPR effects, suppressing tumor growth and increasing the survival rate in various tumor animal models. However, enhanced specificity could be further considered through active targeting nanoliposome [67,68,69]. Although combinational use of internal nanotargeted ^188^Re-liposome with chemotherapy in clinical settings is still in its infancy, it has heralded the dawn of a new era in future cancer treatment through our experimental achievements in preclinical animal models investigation and radio/chemo-combination therapeutic effect and survival comparison results. Nevertheless, two issues need to be overcome in those novel therapeutic strategies’ design. First, the exploration of the optimal protocol of active nanotargeted radionuclide therapy in combination with nanotargeted chemotherapy to decrease the nanotargeted drugs that accumulate in the reticulaendothelial system, such as the liver, spleen, and red marrow. Second, each cancer type possesses specific clinical characteristic properties, such as the tumor microenvironment, and invasion time periods, etc. Increasing preclinical and clinical investigated results showed the effect of radiation in terms of modulating immunity could increase its therapeutic efficacy role in systemic anti-cancer treatment [67]. Local external radiation therapy in combination with chemotherapy of tumor-triggering systemic response is described as an abscopal effect. Our systemic investigations and comparison of nanotargeted ^188^Re-liposome pharmacology as well as the therapeutic effect in various animal models and translational clinical research results point to a new novel formulation of nanotargeted ^188^Re-liposome combining radio/chemo/EBRT/immuno-combination therapy. This could potentially enhance the nuclear imaging quality and accuracy in addition to the systemic tumor lesions treatment effectiveness.

In conclusion, this paper provides an overview and highlights our multifunctional and multi-institutes collaboration of nanotargeted ^188^Re-liposome from preclinical tumor and ascites animal models to preliminary clinical translation studies. The results demonstrate ^188^Re-liposome is both capable and favorable in vivo tumor targeting and nuclear imaging, biodistribution, pharmacokinetics, dosimetry, radiotherapeutic effectiveness, and has a good synergistic radio/chemo-combination effect. However, further efforts and challenges in preclinical and clinical efficacy and toxicity studies are required to translate the new novel nanotargeted ^188^Re-liposome with the radio/chemo/immuno-combination magic bullet therapeutic concept to further advance new technologies to clinical applications for the healthcare benefits of suitable cancer patients.

## Figures and Tables

**Figure 1 ijms-22-03868-f001:**
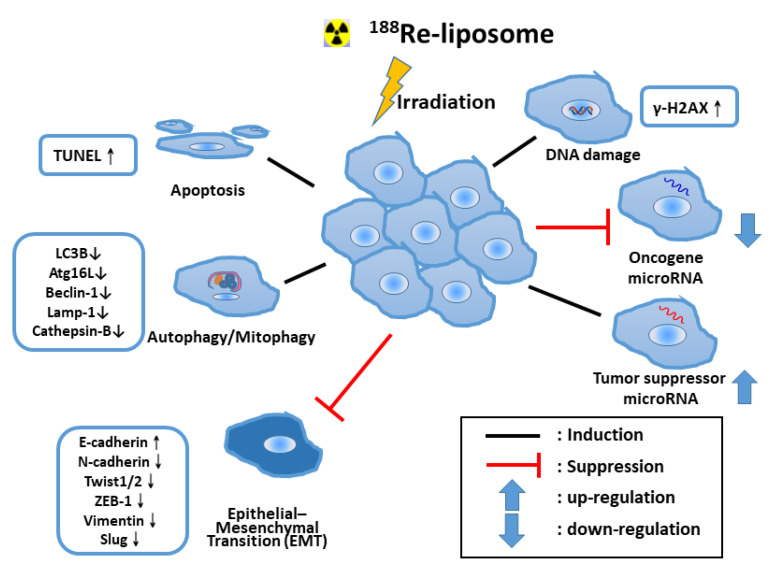
An overview of radiation killing mechanisms of preclinical tumor animal models in internal radiotherapy of nanotargeted ^188^Re-liposome. TUNEL: Terminal deoxynucleotidyl transferase dUTP nick end labeling.

**Figure 2 ijms-22-03868-f002:**
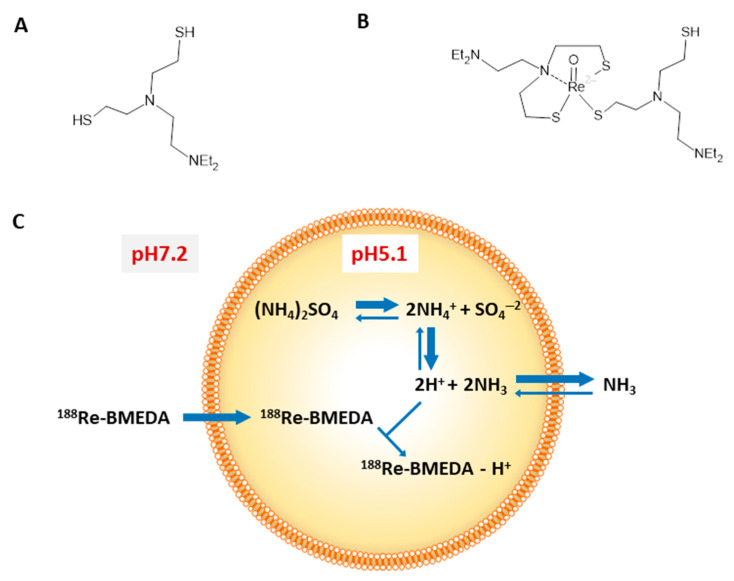
Chemical structure of BMEDA (**A**) and ^188^Re-BMEDA (**B**). Diagram (**C**) depicting after-loading method for liposomes containing ammonium sulfate pH gradient radiolabeled with ^188^Re-BMEDA. The lipophilic form of BMEDA at pH 7.2 crosses the lipid bilayer. Once inside the liposome interior, BMEDA becomes protonated at pH 5.1 and trapped within the hydrophilic liposome interior as ^188^Re-BMEDA-H^+^ form. This research was originally published in JNM [19].

**Figure 3 ijms-22-03868-f003:**
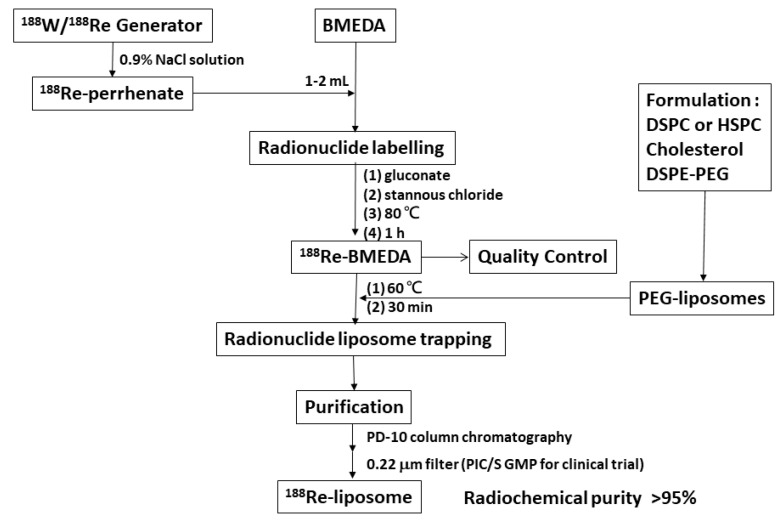
Chemical flowsheet for the preparation of ^188^Re-liposomes. DSPC: 1,2-Distearoyl-sn-Glycero-3-Phosphocholine; HSPC: Hydrogen Soybean Phosphotidylcholine. Figure adapted with permission from [32].

**Figure 4 ijms-22-03868-f004:**
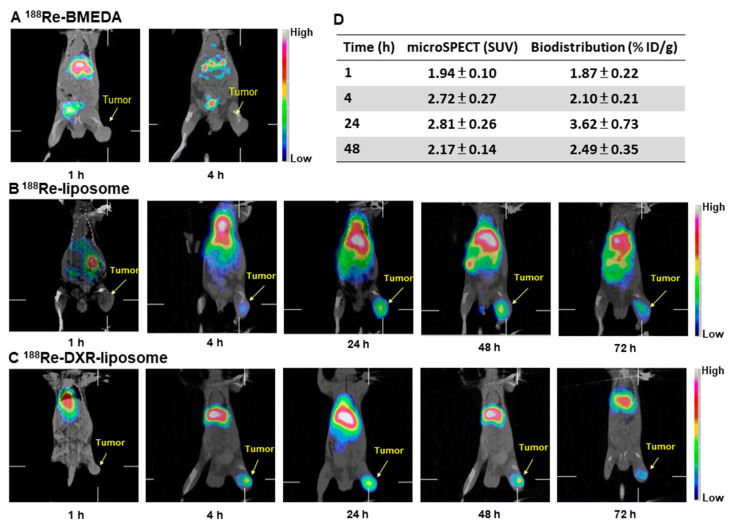
Micro-SPECT/CT imaging of ^188^Re-BMEDA (**A**), ^188^Re-liposome (**B**) and ^188^Re-DXR-liposome (**C**) in C26 tumor-bearing mice. (**D**) Correlation of tumor uptake ^188^Re-liposome analyzed by microSPECT imaging and biodistribution. Arrows indicate positions of subcutaneous tumors. Figures adapted with permission from [30,34].

**Figure 5 ijms-22-03868-f005:**
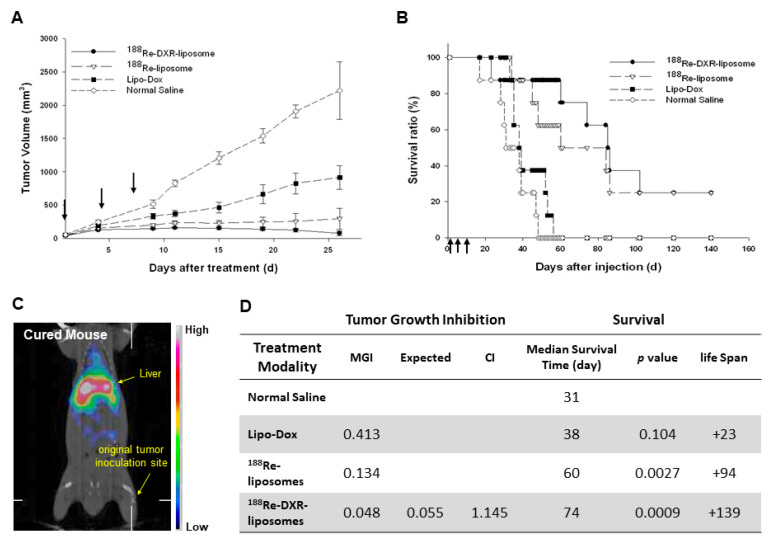
Tumor volume (**A**) and survival ratio (**B**) versus time following administering ^188^Re-DXR-liposome (22.2 MBq of ^188^Re and 2 mg/kg DXR) (●), ^188^Re-liposome (22.2 MBq of ^188^Re) (∇), Lipo-Dox (2 mg/kg DXR) (■) and normal saline (♢) by triple intravenous injection on Day 0, 4 and 8 in C26 murine colon tumor-bearing mice. (**C**) MicroSPECT/CT image of ^188^Re-DXR-liposome at 120 day after treatment. (**D**) Statistics of therapeutic efficacy of ^188^Re-liposome compared with Lipo-DOX. CI: Combination Index; MGI: Mean Growth Inhibition. Figures adapted with permission from [34].

**Figure 6 ijms-22-03868-f006:**
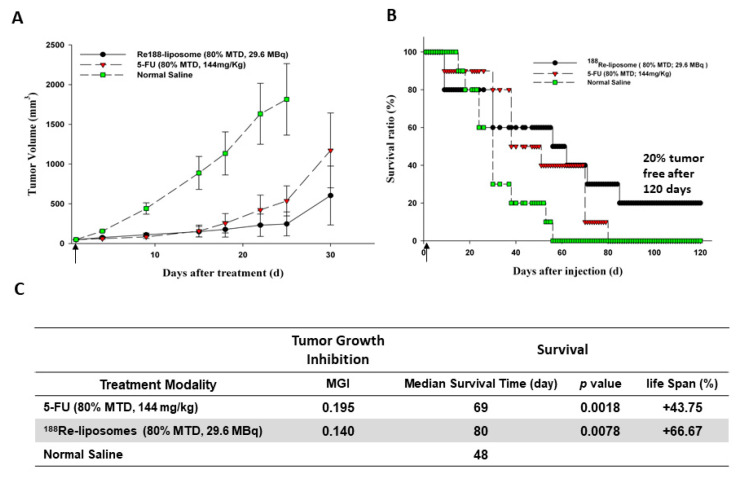
Tumor volume (**A**) and survival ratio (**B**) versus time following administering ^188^Re-liposome (80% MTD, 29.6 MBq) or 5-FU (80% MTD, 144 mg/kg) by single intravenous injection in C26 murine colon tumor-bearing mice. (**C**) Statable 188. Re-liposome compared with 5-FU. Figures adapted with permission from [22].

**Figure 7 ijms-22-03868-f007:**
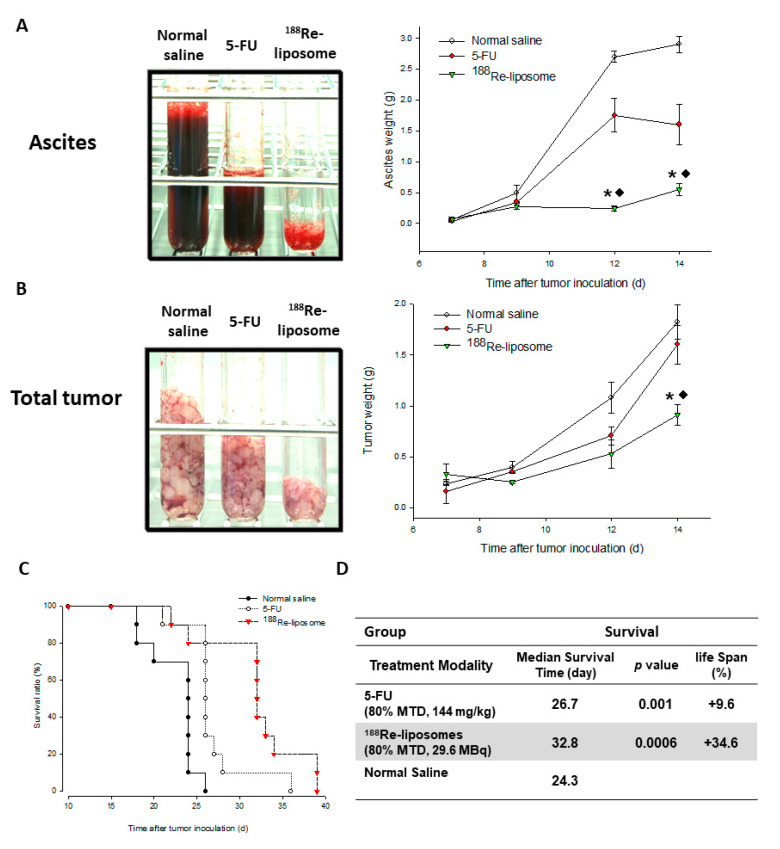
The total ascites weight (**A**), tumors weight (**B**) and survival ratio (**C**) following administering ^188^Re-liposome (80% MTD, 29.6 MBq), 5-FU (80% MTD, 144 mg/kg), and normal saline, respectively by single intravenous injection in C26 peritoneal metastatic tumor-bearing mice. *: significant difference between ^188^Re-liposome- and normal saline-treated groups; ◆: significant difference between ^188^Re-liposome- and 5-FU-treated groups. (**D**) Statistics of therapeutic efficacy of ^188^Re-liposome compared with 5-FU. Figures adapted with permission from [36].

**Figure 8 ijms-22-03868-f008:**
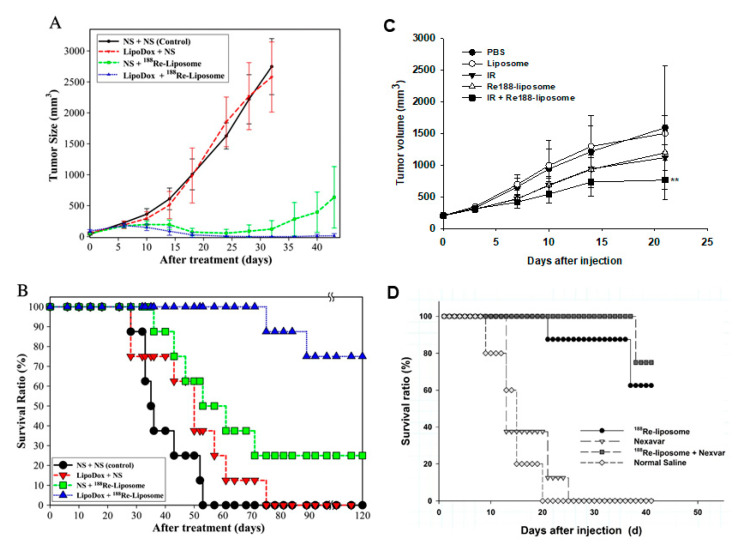
Combination therapy of ^188^Re-liposome with Lipo-Dox, EBRT or Nexavar. Measurements of tumor growth (**A**) and survival ration (**B**) in C26 tumor-bearing mice responded to a treatment regimen in which the mice were injected with either saline or Lipo-Dox (2.5 mg/kg) prior to an injection of the ^188^Re-liposome (22.2 MBq) on day 0. (**C**) Therapeutic effect of EBRT and ^188^Re-liposome. For combination treatment, BE-3 tumor-bearing mice received EBRT (IR, 3 Gy) followed by ^188^Re-liposome (13.2 MBq). Single treatment of EBRT, ^188^Re-liposome, liposome and normal saline were used for comparison. EBRT: external beam radiotherapy; IR: ionizing radiation. (**D**) Therapeutic effect of Nexavar and ^188^Re-liposome. For combination treatment, C26-*luc* murine colon tumor-bearing mice received ^188^Re-liposome (29.6 MBq) and Nexavar (10 mg/kg, once every other day for 1 week) by intrasplenic injection. ^188^Re-liposome or Nexavar treatment only was used for comparison. Figures adapted with permission from [42,43,44].

**Figure 9 ijms-22-03868-f009:**
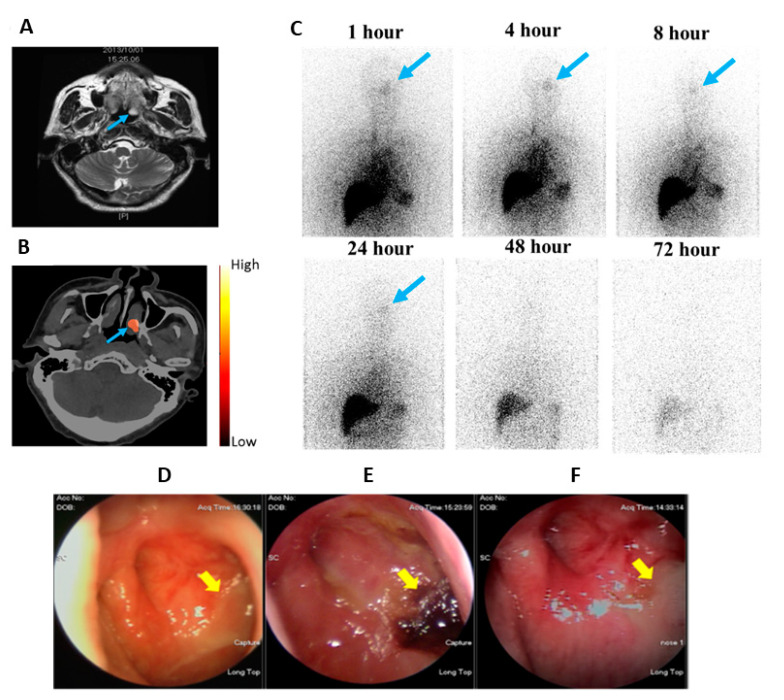
Phase 0 clinical studies of ^188^Re-liposome SPECT/CT imaging showing tumor targeting and uptake in a NPC patient (Pt. 14) with pulmonary and mediastinal metastasis: (**A**) One month before ^188^Re-liposome injection, MRI shows there is a soft tissue lesion (blue arrow) in the left nasopharynx, (**B**) SPECT/CT 24 h after ^188^Re-liposome injection as seen on the MRI, (**C**) Anterior upper-body images at six different time points after ^188^Re-liposome injection (blue arrows indicate the lesion over the left nasopharynx), (**D**) The nasopharyncoscopic examination one month before ^188^Re-liposome injection shows the left nasopharyngeal mass with a crust and mucoid (yellow arrow), (**E**) Two months after ^188^Re-liposome injection, there are engorged blood vessels and irregular surface over the left nasopharynx (yellow arrow) and (**F**) One year after the trial, the nasopharyngoscopic study shows fibrosis over the left nasopharynx (yellow arrow). Figures adapted with permission from [29].

**Table 1 ijms-22-03868-t001:** Summary of preclinical and clinical studies of radio-nanotargeted ^188^Re-(DXR)-liposome.

Preclinical Pharmacology, Therapeutic Effects and Safety Studies				
No	Animal Model	Application	Main Research	Ref
1	C26 murine colon carcinoma	Diagnosis	Solid tumor, drug stability, imaging, PK, Bio-D	[30]
2	HT-29 human colorectal adenocarcinoma	Diagnosis	Solid tumor, drug stability, imaging, PK, Bio-D, T/N ratio, autoradiogram	[31]
3	C26 murine colon carcinoma	Diagnosis	Ascites meta, drug stability imaging, PK, Bio-D, autoradiogram	[32]
4	F98 rat brain glioma	Diagnosis	Solid tumor, drug stability, imaging, PK, Bio-D, high T/N ratio, autoradiogram	[33]
5	C26 murine colon carcinoma	Therapy	Solid tumor, PK, Bio-D, comparison ^188^Re-liposome/Lipo-DOX, synergistic therapy effect of ^188^Re-DXR-liposome, histology	[34]
6	C26 murine colon carcinoma	Therapy	Solid tumor, comparison ^188^Re-liposome/5-FU, safety, radiation effect	[22]
7	C26 murine colon carcinoma	Therapy	Ascites meta, Imaging, PK, Bio-D, autoradiogram, comparison ^188^Re-liposome/Lipo-DOX, synergistic therapy effect of ^188^Re-DXR-liposome, histology	[35]
8	C26 murine colon carcinoma	Therapy	Ascites meta, Imaging, PK, Bio-D, dosimetry, comparison ^188^Re-liposome/5-FU	[36]
9	C26 murine colon carcinoma	Therapy	Lung meta, PK, Bio-D, dosimetry, excretion, comparison ^188^Re-liposome/5-FU	[37]
10	LS-174T human colon adenocarcinoma	Therapy	Soild tumor, imaging, PK, Bio-D, dosimetry, comparison ^188^Re-liposome/5-FU	[38]
11	4T1 murine breast cancer	Therapy	Orthotopic tumor, imaging, Bio-D, comparison ^188^Re-liposome/Lipo-DOX	[39]
12	NCI-H292 non–small cell lung cancer	Therapy	Solid tumor, imaging, Bio-D, PK, comparison ^188^Re-liposome/^188^Re-BMEDA	[40]
13	FaDu human hypopharyngeal carcinoma	Therapy	Orthotopic meta, imaging, Bio-D, PK, comparison ^188^Re-liposome/^188^Re-BMEDA	[27]
14	ES-2-luc human ovarian cancer	Therapy	Ascites meta, imaging, comparison ^188^Re-liposome/^188^Re-BMEDA, radiation effect	[23]
15	F98 rat brain glioma	Therapy	Solid tumor, dosimetry, histology, comparison ^188^Re-liposome/^188^Re-BMEDA	[41]
16	C26 murine colon carcinoma	Therapy	Liver meta, imaging, combination of ^188^Re-liposome/Nexavar	[42]
17	C26 murine colon carcinoma	Therapy	Solid tumor, Bio-D, PK, combination of ^188^Re-liposome/Lipo-Dox	[43]
18	BE-3 human esophageal adenocarcinoma	Therapy	Solid tumor, imaging, Bio-D, safety, combination of ^188^Re-liposome/EBRT	[44]
19	Acute toxicity of BMEDA in ICR mice	Safety	No Observed Adverse Effect Level (3 mg/kg)	[45]
20	Acute toxicity of BMEDA in Beagle dog	Safety	No Observed Adverse Effect Level (2 mg/kg)	[46]
21	Extended acute toxicity of ^188^Re-liposome in SD rat	Safety	No Observed Adverse Effect Level (185 MBq/kg)	[47,48]
**Clinical Safety and Therapeutic Effects Studies**				
1	Phase 0 eIND-metastatic tumors, 12 patients	Safety	Imaging, safety, dosimetry	[29]
2	Low dose clinical trial for 2 ovarian cancer patients	Therapy	Safety, preliminary therapy evaluation, radiation effect	[25]

Bio-D: Biodistribution; BMEDA: *N*,*N*-bis(2-mercaptoethyl)-*N*′,*N*′-diethyl-ethylenediamine; DXR: doxorubicin; EBRT: External beam radiotherapy; eIND: exploratory Investigational New Drug; ICR mice: Institute of Cancer Research mice; PK: Pharmacokinetics; SD rat: Sprague Dawley rat; T/N ratio:Tumor/Non-tumor ratio.

## Data Availability

Not applicable.

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
