# Peer review of "Translating Research for the Radiotheranostics of Nanotargeted 188Re-Liposome"

_ijms, 2021, doi:10.3390/ijms22083868_

Round 1
Reviewer 1 Report
The authors have largely addressed the concerns raised earlier, and I can recommend publication of this review. The editorial office can help check for grammar and sentence flow.
Author Response
Reviewer 1
Comments and Suggestions for Authors
The authors have largely addressed the concerns raised earlier, and I can recommend publication of this review. The editorial office can help check for grammar and sentence flow.
Ans: Thanks for reviewer’s comments and support.
Reviewer 2 Report
This manuscript presents the results of many experimental studies conducted at the facilities where the authors work to which they added a review of the literature. The article has already undergone important and correct changes requested by two other reviewers.
This review is interesting but a few minor points need attention:
I recommend using the: “Consensus nomenclature rules for radiopharmaceutical chemistry
- setting the record straight - (30th April 2017)”. Nucl. Med Biol., 2017 Dec; 55:v-xi.
doi: 10.1016/j.nucmedbio.2017.09.004.Epub 2017 Oct 2. https://pubmed.ncbi.nlm.nih.gov/29074076/
1.Abstract:
To delete: “The phase 0 and preliminary translational clinical study have also been carried out for metastatic tumor patients, showing low dose nano-targeted 188Re-liposome administration achieving favorable tumor accumulation and retention, tumor to normal organ uptake ratios, and tumor response for a subset of cancer patients”
2. Keywords:
Dosimetry (to delete); nanoliposome; liposome (added); nuclear imaging; pharmacokinetics (to delete); radiotherapy; (to delete); radiotheranostics; rhenium-188
3. Introduction:
3.1 At the end of this sentence "Harrington and his colleague reported the biodistribution and imaging of 111In-DTPA-labeled PEGylated liposomes in advanced cancer patients. Tumor images from various cancer types including the breast, head and neck, bronchus, glioma and cervix were obtained by gamma camera and SPECT / CT [16] " referring to the use of PEGylated also cite this more recent bibliography: Boschi, A. et al. PEGylated N-methyl-S-methyl dithiocarbazate as a new reagent for the high-yield preparation of nitride Tc-99m and Re-188 radiopharmaceuticals. (2010) Nuclear Medicine and Biology, 37 (8), pp. 927-934.
3.2 At the end of this sentence: “Recently, with improvements in radiolabeling methodologies development” cite this recent bibliography: Uccelli, L. et al., A. Automated preparation of Re-188 lipiodol for the treatment of hepatocellular carcinoma. (2011) Nuclear Medicine and Biology, 38 (2), pp. 207-213.
-Line 46 pag .2 change “:” after “There are two major advantages of 188Re over other radionuclides in radio-theranostic application “, with “.”
4.Preparation of 188Re.
4.1 Change title with: Rhenium-188 production.
4.2 The paragraph gives little information on the developments that have taken place in recent years on rhenium-188 generators. I suggest to give more information about this point or alternatively add some more references as for example, Journal of Chemistry, Volume 2014, Article ID 529406, 14 pages http://dx.doi.org/10.1155/2014/529406 and . F. Knapp Jr., S. Mirzadeh, M. Garaland, B. Ponsard, and R. Kuznetsov, “Reactor Production and processing of 188W, Production of Long Lived Parent Radionuclides for Generators: 68 Ge, 82 Sr, 90 Sr and 188 W,” IAEA Radioisotopes and Radiophar- maceuticals Series No. 2, 2010.
4.3 move table 1 before paragraph 2
5. Acute toxicity of BMEDA and 188Re-liposome.
Change title with: 4.4 Acute toxicity of BMEDA and 188Re-liposome
6. Therapeutic effects of 188Re-liposome
Change title with: 5 Therapeutic effects of 188Re-liposome
7. Mechanisms of 188Re-liposome anticancer effects.
Authors should integrate this paragraph into the Introduction.
8. Clinical trials of 188Re-liposome
Change title with: 5. Clinical studies of 188Re-liposome
9. Conclusions and Future Directions
Change title with: 6. Conclusion
I would recommend the acceptance of the manuscript after appropriate modifications.
Author Response
Reviewer 2
This manuscript presents the results of many experimental studies conducted at the facilities where the authors work to which they added a review of the literature. The article has already undergone important and correct changes requested by two other reviewers.
This review is interesting but a few minor points need attention:
I recommend using the: “Consensus nomenclature rules for radiopharmaceutical chemistry
- setting the record straight - (30th April 2017)”. Nucl. Med Biol., 2017 Dec; 55:v-xi. doi: 10.1016/j.nucmedbio.2017.09.004.Epub 2017 Oct 2. https://pubmed.ncbi.nlm.nih.gov /29074076/
- Abstract:
To delete: “The phase 0 and preliminary translational clinical study have also been carried out for metastatic tumor patients, showing low dose nano-targeted 188Re-liposome administration achieving favorable tumor accumulation and retention, tumor to normal organ uptake ratios, and tumor response for a subset of cancer patients”
Ans: Thanks for reviewer’s comments and suggestions. We have deleted the sentence.
- Keywords:
Dosimetry (to delete); nanoliposome; liposome (added); nuclear imaging; pharmacokinetics (to delete); radiotherapy; (to delete); radiotheranostics; rhenium-188
Ans: Thanks for reviewer’s comments and suggestions. We have deleted these keywords.
Introduction:
3.1 At the end of this sentence "Harrington and his colleague reported the biodistribution and imaging of 111In-DTPA-labeled PEGylated liposomes in advanced cancer patients. Tumor images from various cancer types including the breast, head and neck, bronchus, glioma and cervix were obtained by gamma camera and SPECT / CT [16] " referring to the use of PEGylated also cite this more recent bibliography: Boschi, A. et al. PEGylated N-methyl-S-methyl dithiocarbazate as a new reagent for the high-yield preparation of nitride Tc-99m and Re-188 radiopharmaceuticals. (2010) Nuclear Medicine and Biology, 37 (8), pp. 927-934.
Ans: Thanks for reviewer’s comments and suggestions. We have added the references as ref 18.
3.2 At the end of this sentence: “Recently, with improvements in radiolabeling methodologies development” cite this recent bibliography: Uccelli, L. et al., A. Automated preparation of Re-188 lipiodol for the treatment of hepatocellular carcinoma. (2011) Nuclear Medicine and Biology, 38 (2), pp. 207-213.
Ans: Thanks for reviewer’s comments and suggestions. We have added the references as ref 17.
-Line 46 page .2 change “:” after “There are two major advantages of 188Re over other radionuclides in radio-theranostic application “, with “.”
Ans: Thanks for reviewer’s comments and suggestions. We have changed the sentence.
4.Preparation of 188Re.
4.1 Change title with: Rhenium-188 production.
Ans: Thanks for reviewer’s comments and suggestions. We have changed the title.
4.2 The paragraph gives little information on the developments that have taken place in recent years on rhenium-188 generators. I suggest to give more information about this point or alternatively add some more references as for example, Journal of Chemistry, Volume 2014, Article ID 529406, 14 pages http://dx.doi.org/10.1155/2014/529406 and . F. Knapp Jr., S. Mirzadeh, M. Garaland, B. Ponsard, and R. Kuznetsov, “Reactor Production and processing of 188W, Production of Long Lived Parent Radionuclides for Generators: 68 Ge, 82 Sr, 90 Sr and 188 W,” IAEA Radioisotopes and Radiophar- maceuticals Series No. 2, 2010.
Ans: Thanks for reviewer’s comments and suggestions. We have added more references as ref 49 and 50.
4.3 move table 1 before paragraph 2
Ans: Thanks for reviewer’s comments and suggestions. We have moved the table before paragraph 2.
- Acute toxicity of BMEDA and 188Re-liposome.
Change title with: 4.4 Acute toxicity of BMEDA and 188Re-liposome
Ans: Thanks for reviewer’s comments and suggestions. We have changed the title.
- Therapeutic effects of 188Re-liposome
Change title with: 4.5 Therapeutic effects of 188Re-liposome
Ans: Thanks for reviewer’s comments and suggestions. We have changed the title
- Mechanisms of 188Re-liposome anticancer effects.
Authors should integrate this paragraph into the Introduction.
Ans: Thanks for reviewer’s comments and suggestions. We have integrated this paragraph into the Introduction.
- Clinical trials of 188Re-liposome
Change title with: 5. Clinical studies of 188Re-liposome
Ans: Thanks for reviewer’s comments and suggestions. We have changed the title.
- Conclusions and Future Directions
Change title with: 6. Conclusion
Ans: Thanks for reviewer’s comments and suggestions. We have changed the title.
I would recommend the acceptance of the manuscript after appropriate modifications.
Ans: Thanks for reviewer’s support.
This manuscript is a resubmission of an earlier submission. The following is a list of the peer review reports and author responses from that submission.
Round 1
Reviewer 1 Report
The manuscript Translating Research for Radio-theranostics of 188Re-liposome, attempts to review recent advances in the translation of 188Re-containing liposomes to the clinic. However, in its current form it fails to do this in a clear and understandable way. The level of English is very poor, making it at times difficult to understand what exactly is the authors are attempting to say. Next to that, a considerable amount of the presented information is not relevant for the manuscript itself. Finally, while practically all of the references concerning 188Re liposomes themselves are papers from one or more of the current authors, in the manuscript it is suggested that the presented studies are from more than one group (e.g. "several investigations have demonstrated that...', line 313). Because of these, and the below mentioned points per chapter, I recommend rejection of this manuscript.
Abstract: consider the audience, some of the abbreviations mentioned are not elaborated upon, but also not widely known. Suggestion to keep it more straightforward and to-the-point.
Introduction: much too long and broad. The manuscript is about translating 188Re liposome research, but the introduction includes too much information, does not lead up to the main point of the review, and e.g. goes into too much irrelevant detail about Vescan (which the authors claim is 'commercially available', but it has in fact been developed but not commercialized). Table 1 does not serve any real purpose, it would be sufficient to just include those references in the introduction as they are not about liposomes at all.
2. Preparation of 188Re: seems like a summary of the chapter in ref [44], though it is not clear that all the information is from here. Furthermore, the development of the generator might be the most important achievement in making 188Re widely available, but definitely is not in using 188Re. The last sentence (line 155 and further) implies that there are only two places where it is possible to obtain 188Re, which is of course not the case.
3. Preparation and Characterization of 188Re-liposome: Too much irrelevant detail. Stick to main methods, and refer to the papers if readers want more detailed information. Line 179 "usually more than 95%', just present it as average with associated uncertainty.
4. Preclinical studies of 188Re liposome: Line 224 suggests that 188Re liposome was taken up for 36% in the tumor, which is definitely not what the article states, in fact it is the max conc in the blood. The relevance of 188Re-BMEDA in its current form is unclear. There are many values given, but it is difficult to make sense out of what is important and what not. Suggestion to the authors: indicate somewhere why it is preferable to have 188Re in liposomes (with generally lower tumor uptake) rather than tumor-specific targeting vectors? In the Dosimetry section, lines 319-332 are all about biodistribution studies, not dosimetry, and the first paragraph is not very relevant in this level of detail.
5. Acute toxicity of 188Re liposome: this section lacks a lot of relevance. Only the very last line (377-380) actually discusses the toxicity
6. Therapeutic effects of 188Re liposome: again the text hardly mentions what the title indicates it should, the therapeutic effect is only really mentioned in lines 394-398. Tables 2 and 3 completely lack uncertainty information in all the data points they offer.
7. Mechanisms of cytotoxic effects of 188Re liposome: first paragraph is very broad, second ties it a bit to 188Re liposome but recommended to completely rewrite and reorder this chapter to make it more clear as well.
Acknowledgements: this is a review paper. This means it should not contain any new experimental results, and from reading the paper I did not get the idea that it does. Therefore I was very surprised to see the authors thanking a number of people for preparing liposomes and performing animal experiments and dosimetry?
Reviewer 2 Report
This manuscript reviews a unique theranostics system, is timely and serves a good purpose to the scientific community. I will recommend its publication with major revisions. My concerns relate to i) minimal use of the visuals, and the review is largely very wordy. It will be good to include some data depicting figures especially in the sections 4-7 related to pharmacokinetics and biodistribution, toxicity, therapeutic effects and mechanism. Figure 2 does not serve any purpose as the synthesis is not described in the text; ii) The tables are very confusing and do not serve any purpose, or provide any additional information to what is said in the text, and are waste in space; iii) The section on current status of clinical trials (section 8) refers to studies done in 2012/2013, and it is necessary to bring these into focus for 2020, or change the title of this section to “Clinical trials…”; iv) Conclusions and Future outlook section should summarize first what has been extensively discussed in the manuscript, what we learnt from it, and then move into how this can be further elaborated into “next step nanotargeted nucleotides”.
Several other minor corrections include:
Use of the word “Evidences” repeatedly, and it should be “Evidence”; Evidence suggests…….etc.
Line 127: Odd sentence: “There are currently applied into various stages of pre-clinical and clinical studies as listed in Table 1.”
Line 142: “…practical especially in some remote areas where is difficult to…”; change to “…where it is difficult to…”.
Line 172: “Figure 2 showed the scheme for the preparation”; change to “Figure 2 shows the scheme….”.
Line 211: “In compared with 99mTc…” change to “In comparison with…..”
Line 214: “There is no significantly difference between…” change to “There is no significant difference…..”.
Line 216: “These evidences indicate the 188Re….” change to “The evidence indicates that the…..”
Line 251: “All biodistribution have shown that 188Re-Liposome could significant accumulation in the tumor, liver, spleen and kidney” change to “Biodistribution studies have shown…..could significantly accumulate in the tumor…..”.
Line 256: “….of nanoliposome uptake has reported in the…” change to “…uptake has been reported in the…”
Line 259: “The mechanism may due to glioma has neoangiogenesis properties” change to “…may be due to…”
Line 287: “Several tumor-bearing mice models have been demonstrated lon-
gitudinal microSPECT/CT could be used as the imaging of tumor…” change to “….mice models have demonstrated that Iongitudinal microSPECT/CT could be used for the imaging……”
Line 315: “…..carcinoma tumor-bearing mice model indicates 188Re-liposome can accumulate…” change to “….indicate…”
Line 343: Odd sentence “Radio-Tox Lab owns the abilities for evaluation”.
Line 361: “However, in compared with the control mice, 3-mg/kg….” change to “However, in comparison with…..”
Line 377: “Evidences suggest the “No Observed Adverse Effect Level (NOAEL)” of…”. What is the purpose of this abbreviation? Change also to “Evidence suggests….”.
Line 385: “….administration, Maxinium tolerance Dose (MTD)….” Why M and D are in capital here? Change Maxinium to maximum.
Line 387: “Body weight dose-dependently decreased is noted when mice treated with 188Re-Liposome” change to “Body weight dose-dependency decrease….mice is treated….”
Line 388: “Evidences indicate there….” Change to “Evidence indicates that there….”
Line 436: Odd sentence “Chemotherapy own the potential to synergize with radiation….”
Line 478: “Evidences from ovarian….” change to “Evidence from ovarian…” The whole sentence is odd.
Line 480: “In the other hand, beta-irradiation-mediated…” change to “On the other hand….”
Line 510: “…make 188Re-liposome to be convenient used in cancer patients.” Change to “…to be conveniently used…..”